# Development of type 2 diabetes and insulin resistance in people with HIV infection: Prevalence, incidence and associated factors

**Göran Bratt[1,2], Johanna Brännström[2,3], Catharina Missalidis[2,4], Thomas Nyström** [1,5] *

**1** Department of Clinical Science and Education, Karolinska Institutet, Stockholm, Sweden, **2** Department of Infectious Diseases/Venhälsan, South Hospital, Stockholm, Sweden, **3** Division of Infection and Dermatology, Department of Medicine Huddinge, Karolinska Institutet, Stockholm, Sweden, **4** Department of Laboratory Medicine, Division of Clinical Microbiology, Karolinska Institutet, Karolinska University Hospital, Stockholm, Sweden, **5** Department of Internal Medicine, South Hospital, Stockholm, Sweden

* thomas.nystrom@ki.se

## Abstract

### Background

Diabetes and insulin resistance is an emerging issue in people with HIV. HIV-related mortality and morbidities have decreased markedly over the last few decades, while co-morbidities including type 2 diabetes (T2D) have increased.

### Setting

This study investigated the incidence of T2D and insulin resistance in a cohort of HIV-patients on effective treatment.

### Methods

Prevalence and baseline predictors of T2D were assessed in a cohort of 570 HIV-positive patients 50 years or older. Patients without diabetes (n = 505) were followed prospectively over a median period of 7.25 year (2012–2020) until T2D development, death or end of the study. T2D was defined as repeated fasting glucose values $\geq$7.0 mmol/L. Insulin resistance was defined as HOMA-IR $\geq$3.0. Predictors of T2D development (HIV-related parameters, lipids, hypertension, central obesity, inflammation, smoking and use of statins) were assessed using logistic regression analysis.

### Results

30% (153/505) had insulin resistance. During follow up (3485 patient-years) 9% (43/505) developed T2D and 7% (36/505) insulin resistance. Thus, at follow up the prevalence of either T2D or insulin resistance was 46% (232/505). T2D incidence was 1.2/100 patient-years. In multivariate analysis, after adjustment for age, T2D development was associated with baseline insulin resistance, hypertriglyceridemia, central obesity and statin treatment, but no HIV-related factors.

**Data Availability Statement:** All relevant data are within the paper and its S1 Dataset.

**Funding:** Physicians against AIDS research fund (Läkare mot AIDS forskningsfond).

**Competing interests:** The authors have declared that no competing interests exist.

## Conclusion

The incidence of T2D in this cohort of patients with well controlled HIV-infection was high. The predictive factors associated with the development of T2D were not unique for HIV positive patients. The findings underline the importance of lifestyle changes in avoidance of T2D in people with HIV.

## Introduction

Mortality and morbidity among people living with human immunodeficiency virus type 1 (PLHIV) have decreased the last two decades [1]. This is mainly due to a continual increase in uptake of highly efficient and long-term safe combined antiretroviral treatment (cART). Thus, co-morbidities will have a greater impact on the long-term health and survival of HIV-patients.

In the ongoing American prospective HIV Outpatient Study (HOPS) the death rate fell from 12.1 to 1.6 deaths per 100 person-years between 1994 and 2017 [2] whereas the proportion of non-AIDS causes of death (cardiovascular, hepatic, pulmonary and non-AIDS associated malignancies) increased [3]. However, the role and impact of life-long cART on the development and progress of significant comorbidities is as yet unclear.

In Sweden over 95% of all diagnosed HIV-patients are on cART and more than 95% of these are virally suppressed [4]. A recently published Swedish cohort study of 4066 PLHIV followed for 15 years found a non-AIDS-associated mortality rate of successfully treated patients to be 2.4 times greater than that of 8072 HIV-negative controls matched for age, gender and region of birth [5].

Many studies have indicated that PLHIV on cART have an over risk for myocardial infarction, cerebrovascular events and type 2 diabetes (T2D) [6–11]. Some investigators have found T2D to be 3–4 times more common among PLHIV on cART as compared to the general population, the prevalence being reported as up to 20% in the 51–60 years age group [12]. T2D among PLHIV is associated with an increased risk for cardiovascular events (e.g. RR 3.0, in the DAD study), cardiac diastolic dysfunction, liver fibrosis (even without hepatitis C co-infection), chronic kidney disease and peripheral neuropathy [13–17]. Moreover, in the general population, insulin resistance, as measured by the homeostasis model assessment for insulin resistance (HOMA-IR) index, has been reported to be an independent risk factor for CVD events and all-cause mortality in subjects with arterial disease even without manifest T2D [18].

The HLA B 5701 allele was recently suggested to protect against type 1 diabetes (T1D) in the large international Type 1 Diabetes Genetics Consortium (T1DGC) study [19]. This allele is also the only genetic marker routinely screened for in HIV-care in order to avoid abacavir hypersensitivity [20]. In addition to indicating abacavir hypersensitivity the HLA B 5701 allele is also associated with restriction of HIV-replication in long-term non-progressors [21]. Whether HLA B 5701 expression also has an impact on T2D development has, to the best of our knowledge, not been investigated.

The aim of this study was to investigate the incidence of T2D and insulin resistance in a group of well controlled PLHIV over 50 years of age and over a long-term period. The decision to only include this age group was based on our clinic´s focus on co-morbidity in an aging HIV-population.

Baseline predictors of T2D such as metabolic and lifestyle related parameters (lipids, hypertension, central obesity, insulin resistance and smoking), inflammation and statin use as well as HIV-related parameters, HLA B 5701 positivity and cART-composition, were documented.

## Material and methods

### Ethics

The study was approved by the Karolinska Institutet Ethics Committee (Regionala Etikprövningsnämden, Karolinska Institutet), Stockholm Sweden (2015 2[th] of September), and written informed consent from the study subjects was obtained.

### Patients

In early 2012 all HIV-patients older than 50 years received written and verbal information about the study and those who gave informed consent verbally were included. Out of 573 patients 570 agreed to participate.

At baseline the 570 eligible patients were analysed cross-sectionally. Of these, one percent (7/570) had T1D and 10% (58/570) had T2D, Fig 1. Of the T2D patients 69% (40/58) were treated with glucose lowering medication: 14 individuals were on insulin only, 20 on metformin only and 6 on insulin in combination with metformin; 31% (18/58) were treated with diet and physical activation only. The remaining 505 patients without diabetes (referred to as "the cohort") were prospectively followed with yearly routine testing in the clinic for the development of the outcomes of interest, i.e. T2D and insulin resistance.

Inclusion criteria for the study were: 1. Having a verified HIV-1 infection being treated at Venhälsan, South Hospital, Stockholm. 2. Born before the 1[st] of January 1963. 3. Had given verbal consent to participate. Exclusion criteria were: Having been diagnosed with T1D or T2D prior to the start of the study and refusing to participate.

### Procedure and follow up

Since the introduction of protease-inhibitor based cART (PI-cART) in 1996 there has been a special interest in metabolic and hemostatic parameters in our clinic [22] and a yearly

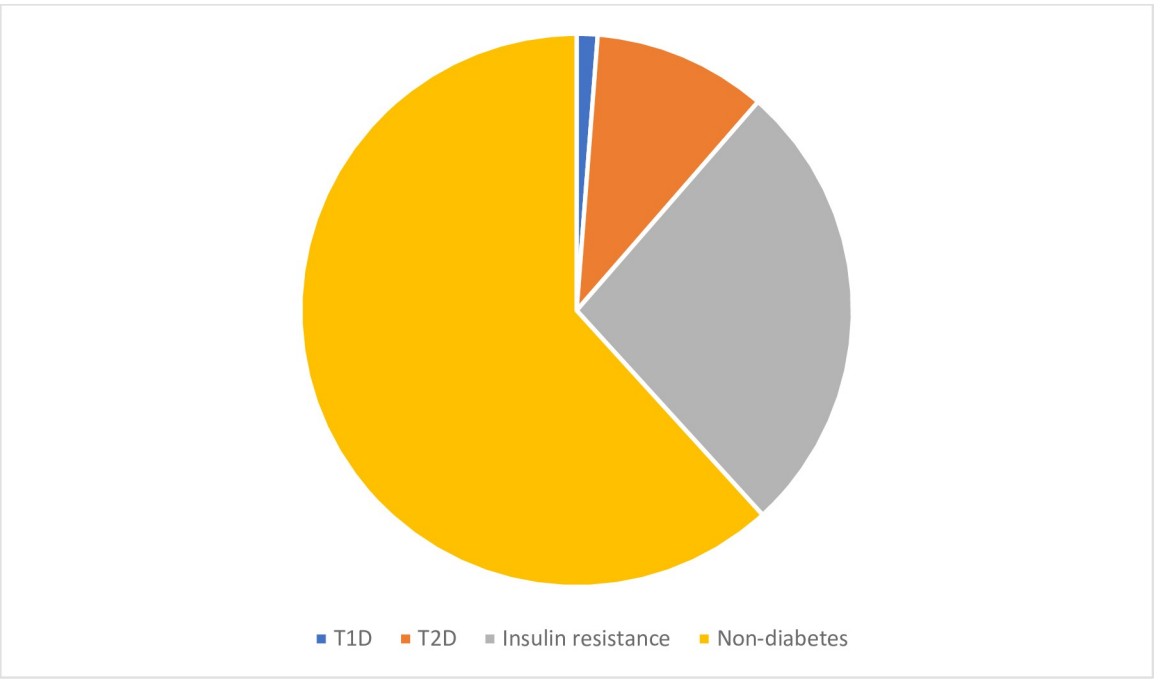

**Fig 1. Proportions of T1D, T2D and insulin resistance in the cross-sectional analysis at baseline (n = 570).**

evaluation of all patients includes the fasting testing of total cholesterol, HDL- and LDL cholesterol, triglycerides, fasting insulin and glucose, and high sensitivity C-reactive Protein (hsCRP). Also, HIV-RNA, CD4, blood pressure, length, weight, BMI, waist and hip circumference and routine hematological, renal and liver tests are documented. All patient data were available for this study and collected from patients´ medical records and assembled in an anonymized database including all results from the yearly evaluation at baseline. Also, date of HIV-diagnosis, HLA B 5701 status, date of cART start, time with known HIV-infection (months), the lowest (nadir) CD4, time (months) with CD4 $< 200$ x$10^6$/L (as defined by the period from the first CD4 value $< 200$ x$10^6$/L up to the first CD4 value $> 200$ x$10^6$/L), the lowest (nadir) CD4/CD8 ratio, the initial HIV-RNA, the highest HIV-RNA, length of any cART interruption longer than 1 month, smoking history, history of treatment with the d-drugs Stavudine (d4t) and Didanosine (DDI) (months), actual CD4 and HIV-RNA and all on-going medication, both cART and medications for co-morbidity, were obtained.

Insulin resistance was defined using the homeostasis model assessment for insulin resistance (HOMA-IR) ((fasting insulin (mIU/l) x fasting glucose (mmol/l))/22.5) in all non-diabetic patients. Although a fasting HOMA-IR provides a valid surrogate marker to assess peripheral insulin sensitivity in epidemiological studies in subjects without diabetes there is no consensus in the literature to define the optimal cut-off value [23, 24].

For the definition of insulin resistance in non-diabetic patients HOMA-IR $\geq 3.0$ was used (24). This value was close to the median of 2.7 (interquartile range: 1,8–4,1) among non-diabetic HIV-patients 50 years or older in our clinic in 2020 (n = 455), and similar to the cut-off value defining insulin resistance in the BRAMS study [25]. T2D was defined as repeated fasting glucose values $\geq 7.0$ mmol/l [26]. Hyperlipidaemia was defined as repeated values of either total cholesterol $\geq 5.0$ mmol/l, LDL cholesterol $\geq 3.0$ mmol/l or on-going lipid lowering therapy. Pathological fasting glucose and lipid values were verified at least once.

Non-HDL cholesterol was calculated as the difference between total cholesterol and HDL cholesterol. Hypertension was defined by repeated BP $\geq 140/90$ mmHg on at least two different occasions or on-going antihypertensive treatment. Central obesity was defined by waist circumference $\geq 94$ cm and $\geq 80$ cm for men and women, respectively.

## Outcomes

The patients in the cohort were followed up from baseline until the last yearly control, loss to follow up or death. All files were re-reviewed regarding the most recent testing for T2D and insulin resistance the 1st of May 2020. The *primary outcome* was a diagnosis of T2D. Also, development of insulin resistance was one outcome of interest.

## Statistical analysis

Descriptive statistics were used to provide an overview of all HIV and metabolic variables. Patient data are presented as median with 95% confidence interval for continuous variables and numbers and percentages for categorical variables. Fishers exact test was used to compare proportions. The significance level was set at p $<0.05$. T2D in the cohort was analysed using univariate logistic regression to obtain odds ratio (OR) for the association between the dependent variable and the independent variables. In the full multivariate logistic regression, all factors with p-values below 0.2 in the univariate analysis were entered and controlled for age. The patients were followed up until moving to another city, death or the last yearly test before 1st of May 2020. The last value was carried forward and used in the analysis. The statistic program SPSS, version 25 (IBM Svenska AB, 16492 Stockholm), was used for the analysis.

## Results

### Description of the prospectively followed cohort

The cohort consisted of 87% (n = 440) males and 13% (n = 65) females. Of the males 89% (n = 391) were Caucasians, 5% (n = 23) of Latin origin, 3% (n = 14) of African descent, 1% (n = 5) from the Middle East and 2% (n = 7) from Asia. Of the females 55% (n = 36) were Caucasians, 8% (n = 5) of Latin origin, 34% (n = 22) of African descent and 3% (n = 2) from Asia.

The majority (98%) were infected through sexual contact: 393 were men who have sex with men (MSM), 35 were heterosexually infected men, 11 were infected through iv drug use. Of the females 62 were sexually infected and 4 through iv drug use.

More than 95% of the patients were on cART and over 95% had HIV-RNA <100 copies/ ml. A previous history of advanced immune deficiency and AIDS had occurred in 50% and 20%, respectively, Table 1. Furthermore, 29% had had a treatment interruption of at least one month. The median time of treatment interruption was 14.5 months (range: 1–105 months).

The CD4 count had increased from less than 200 x10$^6$/L to 600 x10$^6$/L, Table 1.

**Table 1. Descriptive population data of the patients without diabetes (n = 505) at baseline.**

|  | Patients |
|---|---|
| **Total number (M; F, n (%))** | **505 (M 440 (88%); F 65 (12%))** |
| **Median age in years (range)** | **57 (49–83)** |
| **HLA B 5701 pos n; (%)** | **21/500 (4%)** |
| **Median time with known HIV-infection in months (range)** | **204 (6–374)** |
| **AIDS diagnosis n; (%)** | **94 (19%)** |
| **Median initial, pre-ART, HIV-RNA in copies/ml (range)** | **42100 (19–10000000)** |
| Median CD4 nadir x10$^6$/L median (range) | **193 (0–870)** |
| CD4 nadir <200 x10$^6$/L n; (%) | **266 (53%)** |
| **Median time in months without cART (range)** | **54 (0–372)** |
| **Treatment interruption of at least one month: prevalence in %; median time in months (range)** | **26%; 16.0 (1–105)** |
| **Ever on D4t (Stavudine) n; (%)** | **179 (35%)** |
| **Ever on DDI (Didanosine) n; (%)** | **179 (35%)** |
| **On cART n; (%)** | **491 (97%)** |
| **cART including a NNRTI n; (%)** | **306 (61%)** |
| **cART including a Protease inhibitor n; (%)** | **162 (32%)** |
| **cART including an Integrase inhibitor n; (%)** | **100 (20%)** |
| **cART including Emtricitabine n; (%)** | **260 (51%)** |
| **cART including Lamivudine n; (%)** | **193 (38%)** |
| **cART including Abacavir n; (%)** | **176 (35%)** |
| **cART including Tenofovir n; (%)** | **268 (53%)** |
| **cART including Zidovudine n; (%)** | **5 (1%)** |
| Median CD4 count x10$^6$/L (range) | **600 (30–1620)** |
| **HIV-RNA <100 copies/ml at baseline n; (%)** | **489 (97%)** |
| **Hypertension n; (%)** | **205 (41%)** |
| **Hyperlipidemia n; (%)** | **202 (40%)** |
| **Central obesity n; (%)** | **114 (23%)** |
| **Insulin resistance n; (%)** | **153 (30%)** |
| **On statin treatment n; (%)** | **115 (23%)** |

The majority had a non-nucleoside reverse transcriptase inhibitor (NNRTI)-based regimen, one third were on protease inhibitors (PIs) and approximately 20% on an integrase inhibitor. The only integrase inhibitor used at baseline was Raltegravir, Table 1.

Insulin resistance occurred in 30% (153/505), Table 1. Hyperlipidemia and hypertension were common. Treatment with an ACE inhibitor occurred in 17% and with an Angiotensin II antagonist in 8%, Table 1.

### Development of T2D and insulin resistance at follow up

During a median follow-up time of 7.25 years (3485 patient-years) 9% (43/505) (M: n = 43; F: n = 0) developed T2D (incidence of 1.2/100 patient-years). Another 36 patients, 7% (36/505) (M: n = 19; F: n = 17) developed insulin resistance (incidence of 1.0/100 patient-years). Thus, at follow up, 79 patients (M: n = 62; F: n = 17) had developed either T2D or insulin resistance (incidence of 2.3/100 patient-years). In total, the prevalence of either T2D or insulin resistance was 46% (232 (153+79)/505) at follow up, Fig 2.

Death occurred in 8% (42/505) (mortality rate 1.2/100 patient-years).

### Predictors of T2D at follow up

At follow up high triglycerides, central obesity, statin treatment and insulin resistance at baseline were associated to T2D development in univariate and multivariate analysis, controlled for age. Neither HIV-related factors nor HLA B5701 status or a history of Didanosine or Stavudine usage had any significant influence on T2D development, Table 2.

### Characteristics of the HLA B5701 positive patients

At baseline, regarding all patients (n = 570), T2D was more common among HLA B 5701 positive as compared to HLA B5701 negative patients (22% (6/27) vs 10% (52/531); p = 0.05).

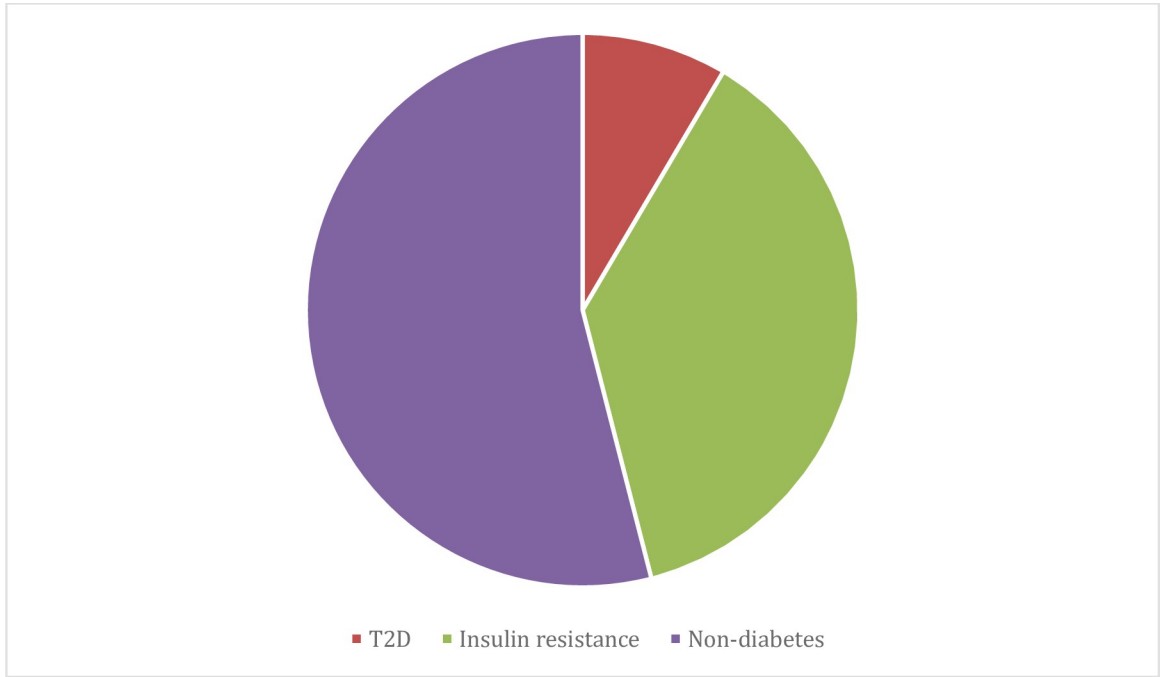

■ T2D   ■ Insulin resistance   ■ Non-diabetes

**Fig 2. Proportions of T1D, T2D and insulin resistance in the prospectively followed cohort at follow up (n = 505).**

**Table 2. HIV-parameters, treatment, metabolic and inflammatory factors at baseline in relation to T2D development during follow up (n = 505).** Uni and multivariate analysis at baseline. The multivariate analysis was controlled for age. All parameters with p<0.20 in the univariate analysis were entered in the multivariate equation. Odds Ratio (OR) and 95% confidence interval.

| | Univariate OR (95% CI) | P-value | Multivariate OR (95% CI) | P-value |
|---|---|---|---|---|
| HLA B 5701 positive | 1.6 (0.5–5.5) | 0.46 | | |
| HIV-months >the median (209 months) | 0.8 (0.4–1.5) | 0.52 | | |
| Ever an AIDS diagnosis | 0.8 (0.3–1.8) | 0.59 | | |
| First HIV-RNA >300000 copies/ml | 0.8 (0.4–1.6) | 0.59 | | |
| CD4 nadir <200 x $10^6$/L | 1.4 (0.7–2.7) | 0.31 | | |
| CD4<200 x $10^6$/L for >12 months | 0.9 (0.4–1.9) | 0.76 | | |
| Ever on D4t (Stavudine) | 0.9 (0.5–1.7) | 0.88 | | |
| Ever on DDI (Didanosine) | 1.2 (0.6–2.3) | 0.60 | | |
| On an NNRTI | 0.9 (0.5–1.6) | 0.68 | | |
| On a protease inhibitor | 1.2 (0.6–2.2) | 0.68 | | |
| On an integrase inhibitor | 0.6 (0.2–1.4) | 0.22 | | |
| On abacavir | 1.5 (0.8–2.7) | 0.25 | | |
| On tenofovir | 0.8 (0.4–1.4) | 0.38 | | |
| Hypertension | 1.9 (1.0–3.6) | 0.051 | | ns |
| Triglycerides >2.6 mmol/L | 3.6 (1.8–7.2) | <0.001 | 2.4 (1.1–5.7) | 0.036 |
| Non-HDL cholesterol >3.8 mmol/L | 1.5 (0.8–2.9) | 0.20 | | ns |
| Central obesity | 2.6 (1.3–5.2) | 0.005 | 2.8 (1.2–5.7) | 0.015 |
| Insulin resistance | 3.4 (1.8–6.5) | <0.0001 | 2.4 (1.1–5.1) | 0.018 |
| Hs CRP >3.0 mg/L | 1.6 (0.8–3.2) | 0.20 | | ns |
| On statin treatment | 2.6 (1.4–4.9) | 0.004 | 2.6 (1.2–5.5) | 0.015 |
| Actual smoker | 0.9 (0.5–1.8) | 0.81 | | |

Expressing HLA B5701 and having been HIV-positive longer in months than the median (HIV-months >209 months) were significantly related to the presence of T2D in univariate analysis. In multivariate analysis, controlled for age, only being HLA B 5701 positive showed a trend (p = 0.055) for association with T2D, Table 3. However, during follow up there was no difference in new T2D diagnoses between HLA B 5701 positive and negative patients (14% (3/21) vs 8% (39/479); ns). None of the patients with T1D were HLA B 5701 positive (0/7).

## Discussion

In this study multiple factors were analysed in PLHIV 50 years or older. T2D and insulin resistance were documented at baseline and during a 7.25 year follow up in a cohort without

**Table 3. Uni and multivariate analysis regarding T2D in all patients including patients with diabetes (n = 570) at baseline.** The multivariate analysis was controlled for age. All parameters with p<0.20 in the univariate analysis were entered in the multivariate equation. OR and 95% confidence interval.

| | Univariate OR (95% CI) | P-value | Multivariate OR (95% CI) | P-value |
|---|---|---|---|---|
| HLA B 5701 positive | 2.6 (1.0–6.8) | 0.046 | 2.6 (1.0–7.1) | 0.055 |
| HIV-months >the median (209 months) | 1.9 (1.1–3.3) | 0.03 | | ns |
| Ever an AIDS diagnosis | 1.7 (1.0–3.1) | 0.11 | | ns |
| First HIV-RNA >300000 copies/ml | 1.4 (0.8–2.5) | 0.21 | | |
| CD4 nadir <200 x $10^6$/L | 1.7 (1.0–3.0) | 0.065 | | ns |
| CD4<200 x $10^6$/L for >12 months | 1.7 (1.0–3.1) | 0.061 | | ns |
| Ever on D4t (Stavudine) | 1.4 (0.8–2.4) | 0.25 | | |
| Ever on DDI (Didanosine) | 1.7 (1.0–2.9) | 0.057 | | ns |
| Ever been a smoker | 1.4 (0.8–2.4) | 0.29 | | |

diabetes. The main findings were that both the prevalence and the incidence of T2D were considerable. At baseline, T2D was about twice as common as in the general Swedish population where the prevalence of T2D requiring treatment was 4.4% in 2013 [27]. During follow up 9% developed T2D, the incidence being 1.2/100 patient years, 3 times higher than in the general Swedish population. We found T2D to develop more often in males than in females, while the development of insulin resistance was equally common.

Well known T2D risk factors, such as insulin resistance, central obesity and hypertriglyceridemia as well as receiving statin treatment but no specific HIV or cART related factors remained predictive of developing T2D after multivariate analysis. Our results parallel those of a similar Italian study where obesity and hypertriglyceridemia were associated with T2D development [28]. In a London study of a similar, but more ethnically diverse group than ours, hypertension and liver steatosis as well as weight gain and longer time with known HIV-infection were found to be predictors of T2D [29].

Comparing prevalence and incidence of T2D in patients with HIV-infection among different international studies is complicated by varying age spans, time of HIV-infection and cART composition. The fairest comparison might be to a Canadian study, similar to ours, which found a T2D incidence of 1.6/100 patient-years, 1.4 times higher than in the general population [30]. In our study the occurrence of T2D at follow up (approximately 18% if the 9% of the baseline population is added to the 9% of the prospectively followed cohort) was higher than in the North American AIDS Cohort Study (MACS) (14%) [31], a recent French study of well-treated patients with an age of 60 or older (14.2%) [32] and a cross-sectional study from the London area (15.1%). In the latter study the prevalence increased from 6.8% over 10 years and contributed to a prevalence of 2.4 times higher than in the general population [29]. On the other hand, our incidence was lower than in both a Spanish cohort study, followed for 2 years after starting cART (2.8/100 patient years) [33], and the MACS cohort (4.7/100 patient years). However, in the MACS study many patients were on first generation protease inhibitors, known to be diabetogenic [31].

Furthermore, our incidence was slightly lower than in a metanalysis which included 44 studies (1.37/100 patient years) [34]. In a Swiss cohort study with over 8000 patients including 2683 patients, 50 years or older, the T2D incidence was age dependent: 0.47/100 person-years in age group 50–64 years which increased to 0.86/100 person-years in those over 64 years [35].

In the current cohort study increased serum lipid levels were common. Lipid-lowering therapy with statins is considered to prevent CVD and decrease all-cause mortality [36, 37]. Statins have also been implicated in having a dose dependent diabetogenic effect [38]. In contrast to the Italian study, we could document an association between statin therapy and T2D development [28]. The diverging results might be explained by differences in prescribing practice and other unknown factors. At our clinic statins are mainly recommended in PLHIV with high cardiovascular risk which could explain the association. In contrast to the Italian study, we failed to find any association between T2D development and historical exposure to Stavudine or Didanosine [28].

The trend suggesting a possible association of T2D with HLA B5701 at baseline was surprising and difficult to explain as HLA B5701 has been found to be protective for T1D [19]. One explanation might be a survival bias since time with HIV-infection was also associated to T2D in univariate regression analyse at baseline and HLA B5701 is associated with restriction of HIV-replication in long-term non-progressors [21]. The HLA B5701 link needs further studying.

With today's well tolerated and safe antiviral drugs, it is uncertain how much, if any, HIV will impair life expectancy in a non-smoking patient with a healthy lifestyle starting cART early after seroconversion. However, the increased T2D risk as well as other factors such as higher levels of inflammatory markers, might impact all-cause mortality [39].

One strength of the current study is the high number of factors that was followed. Another strength is the long term follow up period. However, there are limitations. We have not systematically studied HbA1c levels, since this is not recommended in HIV-infection [11]. Although oral glucose challenge test (OGTT) would give more information about impaired glucose tolerance we consider increased HOMA-index to be an acceptable, simple and patient-friendly surrogate marker for insulin resistance. A HOMA-IR of 3.0 was chosen, -partly from other large studies [23–25], but also due to our own experience. Our finding that patients with insulin resistance have a more than doubled risk to develop T2D strengthens the relevance of HOMA-IR defined insulin resistance. Other weaknesses of the study include the relatively few females and the relatively ethnically homogeneous study group. We also lacked data about weekly physical activity and liver steatosis which would have added valuable information.

A systematic review and meta-analysis in 2008 found an association between PIs and the metabolic syndrome but not T2D [40]. However, the metabolic syndrome or combinations of its components might develop into T2D as our results indicate. In a more recent metanalysis from 2018 the major risk factors, apart from aging, for diabetes and prediabetes were found to be family history of diabetes, Black or Hispanic origin, overweight/obesity, central obesity, lipodystrophy/lipoatrophy, dyslipidemia, metabolic syndrome, increased baseline fasting glycemia, and certain cART regimens [34]. Integrase inhibitors, mainly Dolutegravir, have been anectodically associated to T2D, possibly due to weight gain, which we could not confirm. We failed to find any influence on T2D of different cART regimes or of treatment interruption. The only integrase inhibitor used in our clinic in 2012 was Raltegravir.

In summary, risk factors for T2D in PLHIV are similar to the general population. Our findings underline the importance of focusing PLHIV with central obesity, hypertriglyceridemia, insulin resistance and statin use on lifestyle interventions which can prevent or postpone manifest T2D and could be used as strong motivators in well treated HIV-infection. Recently the European guidelines for cardiovascular prevention were updated with focus on: No tobacco, low intake of saturated fat and high intake of full grain products, vegetables, fruit, fish and regular physical activity and to aim at attaining normal body measurements and blood pressure [41]. Continuous studies of metabolic parameters in well treated PLHIV are important.

## Supporting information

**S1 Dataset.**
(XLSX)

## Acknowledgments

We thank all the patients who participated in this project and the personal at Venhälsan, Södersjukhuset.

## Author Contributions

**Conceptualization:** Göran Bratt, Johanna Brännström, Catharina Missalidis, Thomas Nyström.

**Data curation:** Göran Bratt.

**Formal analysis:** Johanna Brännström, Thomas Nyström.

**Investigation:** Göran Bratt, Johanna Brännström, Catharina Missalidis.

**Methodology:** Thomas Nyström.

**Software:** Göran Bratt.

**Supervision:** Thomas Nyström.

**Validation:** Göran Bratt, Catharina Missalidis.

**Writing – original draft:** Göran Bratt.

**Writing – review & editing:** Göran Bratt, Johanna Brännström, Catharina Missalidis, Thomas Nyström.

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
