## [Decision Letter · Decision Letter 0]

31 Mar 2021

PONE-D-20-38599

Development of type 2 diabetes and insulin resistance in people with HIV infection: Prevalence, incidence and associated factors

PLOS ONE

Dear Dr. Nystrom,

Thank you for submitting your manuscript to PLOS ONE. After careful consideration, we feel that it has merit but does not fully meet PLOS ONE’s publication criteria as it currently stands. Therefore, we invite you to submit a revised version of the manuscript that addresses the points raised during the review process.

Please, take into account all the pertinent comments raised by both reviewers, who have several concerns regarding:

Methodology and patients' cohortsPresentation of the results and conclusions drawn from the dataGrammatical and typographical errors

We look forward to receiving your revised manuscript.

Kind regards,

Graciela Andrei

Academic Editor

PLOS ONE

Journal Requirements:

Thank you for including your ethics statement:  "The study was approved by the regional Ethics committee. ".  

Please provide additional details regarding participant consent. In the ethics statement in the Methods and online submission information, please ensure that you have specified what type you obtained (for instance, written or verbal, and if verbal, how it was documented and witnessed). If your study included minors, state whether you obtained consent from parents or guardians. If the need for consent was waived by the ethics committee, please include this information.

For more information on PLOS ONE's expectations for statistical reporting, please see https://journals.plos.org/plosone/s/submission-guidelines.#loc-statistical-reporting. Please update your Methods and Results sections accordingly.

We note that you have indicated that data from this study are available upon request. PLOS only allows data to be available upon request if there are legal or ethical restrictions on sharing data publicly. For information on unacceptable data access restrictions, please see http://journals.plos.org/plosone/s/data-availability#loc-unacceptable-data-access-restrictions.

5a) If there are ethical or legal restrictions on sharing a de-identified data set, please explain them in detail (e.g., data contain potentially identifying or sensitive patient information) and who has imposed them (e.g., an ethics committee). Please also provide contact information for a data access committee, ethics committee, or other institutional body to which data requests may be sent.

5b) If there are no restrictions, please upload the minimal anonymized data set necessary to replicate your study findings as either Supporting Information files or to a stable, public repository and provide us with the relevant URLs, DOIs, or accession numbers. Please see http://www.bmj.com/content/340/bmj.c181.long for guidelines on how to de-identify and prepare clinical data for publication. For a list of acceptable repositories, please see http://journals.plos.org/plosone/s/data-availability#loc-recommended-repositories.

We note that you have included the phrase “data not shown” in your manuscript. Unfortunately, this does not meet our data sharing requirements. PLOS does not permit references to inaccessible data. We require that authors provide all relevant data within the paper, Supporting Information files, or in an acceptable, public repository. Please add a citation to support this phrase or upload the data that corresponds with these findings to a stable repository (such as Figshare or Dryad) and provide and URLs, DOIs, or accession numbers that may be used to access these data. Or, if the data are not a core part of the research being presented in your study, we ask that you remove the phrase that refers to these data.

Reviewers' comments:

Reviewer's Responses to Questions

**Comments to the Author**

1. Is the manuscript technically sound, and do the data support the conclusions?

Reviewer #1: Partly

Reviewer #2: Yes

2. Has the statistical analysis been performed appropriately and rigorously? 

Reviewer #1: Yes

Reviewer #2: Yes

3. Have the authors made all data underlying the findings in their manuscript fully available?

Reviewer #1: Yes

Reviewer #2: Yes

4. Is the manuscript presented in an intelligible fashion and written in standard English?

Reviewer #1: No

Reviewer #2: Yes

5. Review Comments to the Author

Reviewer #1: Bratt and colleagues investigate the incidence of T2D in PLHIV. Importantly, though morbidity and mortality have decreased significantly over the past few decades in this population, primarily due to improved treatment options, the prevalence of co-morbidities such as T2D is increasing and having a significant negative impact on long-term health. The authors suggest that the incidence of T2D in this cohort of PLHIV is higher than would be expected in the general population and associated risk factors are similar to those in the general population and not related to HIV infection or treatment.

Introduction: 1. The inclusion of HLA B 5701 as part of the 4th paragraph seems random. There is not enough information provided as to its relevance and expanding on why this marker was chosen would provide clarity; perhaps as a paragraph of its own. This is especially relevant since HLA B 5701 is a focus of the results and discussion.

Material and Methods: 1. Remove the additional words 'at baseline' in this title since the longitudinal aspect of the study is described as well.

2. Explain why reference 20 (Koppel et al.) follows the first sentence under the title 'Procedure and follow up.'

3. Provide evidence that a fasting HOMA-IR of ≥3.0 is diagnostic of insulin resistance in PLHIV.

4. In the last sentence of this section, it states that a BMI≥30 kg/m2 was used to define central adiposity. BMI is not a measure of central adiposity.

5. The term Total Clinic Population is used along with Total Study Population to describe the same group. Only one title should be presented.

Results: 1. It would be my suggestion to remove the data presented in the results section on the Total Study Population since the focus of the article is on the Prospective Cohort. Inclusion of the data on the total group makes the results more confusing without adding to the findings. Stating that the initial Total Study Population presented with n=7 with T1D and n=58 with T2D is the only information that is needed. All other data presented should be only on the Prospective Cohort, pre and post follow-up.

2. It states in methods that there were n=570 in the Total Study Population and that the 505 individuals without a diagnosis of diabetes were prospectively followed. If that is the case, please clarify the central adiposity numbers presented in table 1. There is only an n=65 difference between the two groups in total numbers.

3. The relevance of table 4 isn't clear. There needs to be more development of the importance of HLA B 5701 in the introduction.

Discussion: 1. The findings suggest that predictive factors underlying the development of T2D in PLHIV are similar to the general population and not related to the HIV infection itself. There needs to be a paragraph added to the discussion regarding this finding since there are a number of other studies suggesting that HIV-related factors do contribute to the increased prevalence of T2D in PLHIV.

There are a number of spelling and grammatical errors that need to be fixed throughout the manuscript.

Reviewer #2: Please see attached document with same review- easier readability.

The manuscript by Bratt et al. demonstrates a nice prospective study to evaluate the incidence of type 2 diabetes (T2D) and insulin resistance among a cohort of 570 people living with HIV. While the topic is not novel, this study provides additive data to ongoing literature describing prevalence and incidence of T2D in people living with HIV in different countries. The following are suggested changes to add clarity and rigor to the manuscript.

Major Changes:

- The ‘total study population’ is described as a cohort of 570 patients including those with T2D and T1D at baseline and the ‘prospective study population’ as a cohort of 505 patients excluding those with T2D and T1D at baseline.

o It is unclear to me why the authors compare these 2 cohorts when the primary outcome is incidence of T2D. I recommend the authors either better justify why they are including those with baseline diabetes (both T1D and T2D) in the analyses or exclude the ‘total study population’ in the analyses.

Minor Changes:

- Methods section: Please include the inclusion/exclusion criteria

o Justify why you only include those >50 years old.

- Procedure and follow up section: Please be explicit in how many ‘repeated’ measures of fasting glucose, cholesterol, BP measurements, etc were needed or done before diagnosing participants with T2D, hyperlipidemia, HTN, etc.

- Page 7, ‘Description of the population data at baseline’ section: Paragraph that starts with “more than 95% of the patients were on cART…. The second sentence uses the word “respectively”- but it is unclear which groups you are ‘respectively’ referring to. Please clarify.

- ‘Prevalence of T1D, T2D and insulin resistance at baseline and follow up’ section: 2nd sentence needs clarification- what follow up are you referring to when describing the 10% of T2D diagnosed- the 58 patients with T2D were those with T2D at baseline, correct?

- Same section: sentence that begins: “During the follow up 69% (40/58) of the patients diagnosed with T2D…” – the numbers you report do not add up to 100%. I believe the “21%” should be 31%- please check the math.

- Same section: Last sentence of this section that begins with “Death occurs in 8%...”- needs grammatical attention.

- Discussion section: Need more detail on the association of T2D and HLA B5701- compare with other literature.

- Generally: I advise the authors work with a writing coach or copyeditor to improve the flow and readability of the text. There are minor grammatical edits that should be addressed throughout the paper.

6. PLOS authors have the option to publish the peer review history of their article (what does this mean?). If published, this will include your full peer review and any attached files.

Reviewer #1: No

Reviewer #2: No

---

## [Author Response · Author response to Decision Letter 0]

24 Apr 2021

Responses to the reviewers’ specific comments:

Reviewer # 1:

Response: We thank this reviewer for constructive comments on the manuscript.

Re: Introduction

1. “The inclusion of HLA B 5701 as part of the 4th paragraph seems random. There is not enough information provided as to its relevance and expanding on why this marker was chosen would provide clarity, perhaps as a paragraph of its own. This is especially relevant since HLA B 5701 is a focus of the results and discussion”

Response: Thanks for this important comment. We have added more information about, and why, HLA B 5701 was included in the study. To our knowledge this has not been previously examined. We have now inserted following text in the revised manuscript (Introduction): “The HLA B 5701 allele was recently suggested to protect against type 1 diabetes (T1D) in the large international Type 1 Diabetes Genetics Consortium (T1DGC) study [19]. This allele is also the only genetic marker routinely screened for in HIV-care in order to avoid abacavir hypersensitivity [20] . In addition to indicating abacavir hypersensitivity the HLA B 5701 allele is also associated with restriction of HIV-replication in long-term non-progressors [21]. Whether HLA B 5701 expression also has an impact on T2D development has, to the best of our knowledge, not been investigated.” 

Re: Material and Methods

1.”Remove the additional words 'at baseline' in this title since the longitudinal aspect of the study is described as well”

Response: The words 'at baseline' has been removed and the title is now “Material and Methods”

2.”Explain why the ref 20 (Koppel K et al) as part of the 4th paragraph seems random”.

Response: Reference 20 (Koppel et al.) has been moved to a different place in the text (Procedure and follow up): “Since the introduction of protease-inhibitor based cART (PI-cART) in 1996 there has been a special interest in metabolic and hemostatic parameters in our clinic [20] and a yearly evaluation of all patients includes the fasting testing…”

3. “Provide evidence that a fasting HOMA-IR >=3.0 is diagnostic of insulin resistance in PLHIV”. 

Response: This point is well taken. We have included more text motivating this cut off level and inserted two more references (24, 25) to justify cut off value of 3 and above. Also, our own results and clinical experience from cross-sectional HOMA examinations in 455 50+, HIV+ patients were contributed to this value. Following text has been inserted in the revised manuscript (Procedure and follow up): “For the definition of insulin resistance in non-diabetic patients HOMA-IR ≥3.0 was used (24). This value was close to the median of 2.7 (interquartile range: 1,8-4,1) among non-diabetic HIV-patients 50 years or older in our clinic in 2020 (n=455), and similar to the cut-off value defining insulin resistance in the BRAMS study [25]” 

4. “In the last sentence of this section, it states that a BMI≥30 kg/m2 was used to define central adiposity. BMI is not a measure of central adiposity”. Response: We thank for this comment. BMI≥30 kg/m2 has been omitted as a measure of central adiposity (Procedure and follow up): “Central obesity was defined by waist circumference ≥94 cm and ≥80 cm for men and women, respectively”.

5. “The term Total Clinic Population is used along with Total Study Population to describe the same group. Only one title should be presented”. 

Response: Many thanks for this comment. In the revised manuscript the term Total Clinic Population has been omitted

Re: Results: 

1. “It would be my suggestion to remove the data presented in the results section on the Total Study Population since the focus of the article is on the Prospective Cohort. Inclusion of the data on the total group makes the results more confusing without adding to the findings. Stating that the initial Total Study Population presented with n=7 with T1D and n=58 with T2D is the only information that is needed. All other data presented should be only on the Prospective Cohort, pre and post follow-up.”

Response: Again, many thanks for helping us in a more straightforward way show our data. We have, in the revised manuscript, removed the data on Total Study Population from Table 1 and now only include the data for the Prospectively Followed Cohort. The paragraph has been changed to read (Patients [Material and Methods]): “At baseline the 570 eligible patients were analysed cross-sectionally. Of these, one percent (7/570) had T1D and 10% (58/570) had T2D. Of the T2D patients 69% (40/58) were treated with glucose lowering medication: 14 individuals were on insulin only, 20 on metformin only and 6 on insulin in combination with metformin; 31% (18/58) were treated with diet and physical activation only. The remaining 505 patients without diabetes (referred to as “the cohort”) were prospectively followed with yearly routine testing in the clinic for the development of the outcomes of interest, i.e. T2D and insulin resistance”. 

Table 1. Descriptive population data of the patients without diabetes (n=505) at baseline

 Patients

N (M; F, n (%)) 505 (M 440 (88%); F 65 (12%))

Age (median (range); years 57 (49-83)

HLA B 5701 pos (n; (%)) 21/500 (4%)

Time with known HIV-infection 

(months; median (range)) 204 (6-374)

AIDS diagnosis (n; (%)) 94 (19%)

Initial, pre-ART, HIV-RNA

(median (range); copies/ml) 42100 (19-10000000)

CD4 nadir (x106/L; median; (range)) 193 (0-870)

CD4 nadir <200 x106/L (n; (%)) 266 (53%)

Time without cART (median (range); months) 54 (0-372)

Treatment interruption of at least one month: prevalence (%); time (median (range); months) 26%; 16.0 (1-105)

Ever on D4t (Stavudine) (n; (%)) 179 (35%)

Ever on DDI (Didanosine) (n; (%)) 179 (35%)

On cART (n; (%)) 491 (97%)

cART including a NNRTI (n; (%)) 306 (61%)

cART including a Protease inhibitor (n; (%)) 162 (32%)

cART including an Integrase inhibitor (n; (%)) 100 (20%)

cART including Emtricitabine (n; (%)) 260 (51%)

cART including Lamivudine (n; (%)) 193 (38%)

cART including Abacavir (n; (%)) 176 (35%)

cART including Tenofovir (n; (%)) 268 (53%)

cART including Zidovudine (n; (%)) 5 (1%)

CD4 count (median; (range)) 600 (30–1620)

HIV-RNA <100 copies/ml at baseline (n; (%)) 489 (97%)

Hypertension (n; (%)) 205 (41%)

Hyperlipidemia (n; (%)) 202 (40%)

Central obesity 114 (23%)

Insulin resistance (n; (%)) 153 (30%)

On statin treatment (n; (%)) 115 (23%)

2. “It states in methods that there were n=570 in the Total Study Population and that the 505 individuals without a diagnosis of diabetes were prospectively followed. If that is the case, please clarify the central adiposity numbers presented in table 1. There is only an n=65 difference between the two groups in total numbers”. 

Response: Table 1 now only contains data on the Prospectively followed cohort. The incorrect figures for central obesity have been corrected (Table 1 above) 

 3. “The relevance of table 4 isn't clear. There needs to be more development of the importance of HLA B 5701 in the introduction”. 

Response: Table 4 has been removed. In the introduction we have included further motivation of why we included HLA B5701 in the study: See response #1

We have also inserted new text (which replace Table 4) in the revised manuscript for the association between HLA B5701 at baseline and at follow up (Characteristics of the HLA B5701 positive patients [Results]: “At baseline, regarding all patients (n=570), T2D was more common among HLA B 5701 positive as compared to HLA B5701 negative patients (22% (6/27) vs 10% (52/531); p=0.05). Expressing HLA B5701 and having been HIV-positive longer in months than the median (HIV-months >209 months) were significantly related to the presence of T2D in univariate analysis. In multivariate analysis, controlled for age, only being HLA B 5701 positive showed a trend (p=0.055) for association with T2D, Table 2. However, during follow up there was no difference in new T2D diagnoses between HLA B 5701 positive and negative patients (14% (3/21) vs 8% (39/479); ns). None of the patients with T1D were HLA B 5701 positive (0/7).”

Re: Discussion: 

1. “The findings suggest that predictive factors underlying the development of T2D in PLHIV are similar to the general population and not related to the HIV infection itself. There needs to be a paragraph added to the discussion regarding this finding since there are a number of other studies suggesting that HIV-related factors do contribute to the increased prevalence of T2D in PLHIV.”

Response: Thanks for this point. We have added, in the revised manuscript, a paragraph and two more references from 2018 (Nausseau JR and Echecopai-Sabogal J) regarding HIV-related factors that contribute to the increased prevalence of T2D in PLHIV found in other studies (Discussion): “A systematic review and meta-analysis in 2008 found an association between PIs and the metabolic syndrome but not T2D [40]. However, the metabolic syndrome or combinations of its components might develop into T2D as our results indicate. In a more recent metanalysis from 2018 the major risk factors, apart from aging, for diabetes and prediabetes were found to be family history of diabetes, Black or Hispanic origin, overweight/obesity, central obesity, lipodystrophy/lipoatrophy, dyslipidemia, metabolic syndrome, increased baseline fasting glycemia, and certain cART regimens [34]. Integrase inhibitors, mainly Dolutegravir, have been anectodically associated to T2D, possibly due to weight gain, which we could not confirm. We failed to find any influence on T2D of different cART regimes or of treatment interruption. The only integrase inhibitor used in our clinic in 2012 was Raltegravir.

The manuscript has thoroughly been revised regarding spelling and grammatical errors by a native English speaker. 

Reviewer #2: 

We would like to thank this reviewer for valuable critics which has strengthen the manuscript. 

Re: Major Changes:

- The ‘total study population’ is described as a cohort of 570 patients including those with T2D and T1D at baseline and the ‘prospective study population’ as a cohort of 505 patients excluding those with T2D and T1D at baseline.

o It is unclear to me why the authors compare these 2 cohorts when the primary outcome is incidence of T2D. I recommend the authors either better justify why they are including those with baseline diabetes (both T1D and T2D) in the analyses or exclude the ‘total study population’ in the analyses.

Response: This point is well taken (which also was raised by reviewer #1 issue #5). We thank for the advice to rearrange our cohort. We have, in the revised manuscript, removed the data on Total Study Population from Table 1 and now only include the data for the Prospectively Followed Cohort. The paragraph has been changed to read (Patients [Material and Methods]): “At baseline the 570 eligible patients were analysed cross-sectionally. Of these, one percent (7/570) had T1D and 10% (58/570) had T2D. Of the T2D patients 69% (40/58) were treated with glucose lowering medication: 14 individuals were on insulin only, 20 on metformin only and 6 on insulin in combination with metformin; 31% (18/58) were treated with diet and physical activation only. The remaining 505 patients without diabetes (referred to as “the cohort”) were prospectively followed with yearly routine testing in the clinic for the development of the outcomes of interest, i.e. T2D and insulin resistance”. 

Table 1. Descriptive population data of the patients without diabetes (n=505) at baseline

 Patients

N (M; F, n (%)) 505 (M 440 (88%); F 65 (12%))

Age (median (range); years 57 (49-83)

HLA B 5701 pos (n; (%)) 21/500 (4%)

Time with known HIV-infection 

(months; median (range)) 204 (6-374)

AIDS diagnosis (n; (%)) 94 (19%)

Initial, pre-ART, HIV-RNA

(median (range); copies/ml) 42100 (19-10000000)

CD4 nadir (x106/L; median; (range)) 193 (0-870)

CD4 nadir <200 x106/L (n; (%)) 266 (53%)

Time without cART (median (range); months) 54 (0-372)

Treatment interruption of at least one month: prevalence (%); time (median (range); months) 26%; 16.0 (1-105)

Ever on D4t (Stavudine) (n; (%)) 179 (35%)

Ever on DDI (Didanosine) (n; (%)) 179 (35%)

On cART (n; (%)) 491 (97%)

cART including a NNRTI (n; (%)) 306 (61%)

cART including a Protease inhibitor (n; (%)) 162 (32%)

cART including an Integrase inhibitor (n; (%)) 100 (20%)

cART including Emtricitabine (n; (%)) 260 (51%)

cART including Lamivudine (n; (%)) 193 (38%)

cART including Abacavir (n; (%)) 176 (35%)

cART including Tenofovir (n; (%)) 268 (53%)

cART including Zidovudine (n; (%)) 5 (1%)

CD4 count (median; (range)) 600 (30–1620)

HIV-RNA <100 copies/ml at baseline (n; (%)) 489 (97%)

Hypertension (n; (%)) 205 (41%)

Hyperlipidemia (n; (%)) 202 (40%)

Central obesity 114 (23%)

Insulin resistance (n; (%)) 153 (30%)

On statin treatment (n; (%)) 115 (23%)

Re: Minor Changes

Re: Methods section: 

Please include the inclusion/exclusion criteria 

Response: We have now included inclusion and exclusion criteria in the revised manuscript (Patients [Material and Methods]:“Inclusion criteria for the study were: 1. Having a verified HIV-1 infection being treated at Venhälsan, South Hospital, Stockholm. 2. Born 2012 or earlier. 3. Had given verbal consent to participate.

Exclusion criteria were: Having been diagnosed with T1D or T2D prior to the start of the study and refusing to participate”

Justify why you only include those >50 years old. 

Response: We have now in the revised manuscript explained the reason for only including >50 years old patients (Introduction): “The aim of this study was to investigate the incidence of T2D and insulin resistance in a group of well controlled PLHIV over 50 years of age and over a long-term period. The decision to only include this age group was based on our clinic´s focus on co-morbidity in an aging HIV-population.”

Re: Procedure and follow up section: 

Please be explicit in how many ‘repeated measurements of fasting glucose, cholesterol, BP etc were needed or done before diagnosing participants with T2D, hyperlipidemia, HTN, etc.

Response: This point is well taken. We have in the revised manuscript now clarified how many ‘repeat measurements of fasting glucose, cholesterol and BP measurements that were carried out before diagnosing participants with T2D, hyperlipidemia and hypertension (Procedure and follow up): ”T2D was defined as repeated fasting glucose values ≥7.0 mmol/l [26]. Hyperlipidaemia was defined as repeated values of either total cholesterol ≥5.0 mmol/l, LDL cholesterol ≥3.0 mmol/l or on-going lipid lowering therapy. Pathological fasting glucose and lipid values were verified at least once. Hypertension was defined by repeated BP ≥140/90 mmHg on at least two different occasions or on-going antihypertensive treatment”.

- Page 7, ‘Description of the population data at baseline’ section: Paragraph that starts with “more than 95% of the patients were on cART…. The second sentence uses the word “respectively”- but it is unclear which groups you are ‘respectively’ referring to. Please clarify.

Response: Thanks for this comment. The word “respectively”- has been excluded and the sentence rewritten (Description of the prospectively followed cohort): “More than 95% of the patients were on cART and over 95% had HIV-RNA <100 copies/ml. A previous history of advanced immune deficiency and AIDS had occurred in 50% and 20%, respectively, Table 1. Furthermore, 29% had had a treatment interruption of at least one month. The median time of treatment interruption was 14.5 months (range: 1-105 months). The CD4 count had increased from less than 200 x106/L to 600 x106/L, Table 1”.

- The prevalence of T1D, T2D and insulin resistance at baseline and follow up’ section: 2nd sentence needs clarification- what follow up are you referring to when describing the 10% of T2D diagnosed- the 58 patients with T2D were those with T2D at baseline, correct? ‘ 

Response: Thanks for this comment. It is correct that the 58 patients with T2D are those with T2D at baseline. However, this has now been clarified that this 570 was analysed cross-sectional due to HLA B5701 and excluded in the prospectively followed cohort. Following text has been inserted to further clarify this (Patients [Materials and Methods]):“At baseline the 570 eligible patients were analysed cross-sectionally. Of these, one percent (7/570) had T1D and 10% (58/570) had T2D. Of the T2D patients 69% (40/58) were treated with glucose lowering medication: 14 individuals were on insulin only, 20 on metformin only and 6 on insulin in combination with metformin; 31% (18/58) were treated with diet and physical activation only. The remaining 505 patients without diabetes (referred to as “the cohort”) were prospectively followed with yearly routine testing in the clinic for the development of the outcomes of interest, i.e. T2D and insulin resistance.” 

- Same section: sentence that begins: “During the follow up 69% (40/58) of the patients diagnosed with T2D…” – the numbers you report do not add up to 100%. I believe the “21%” should be 31%- please check the math.

Response: We have checked the figures and “21%” should be “31%”. This has now been corrected: “Of the T2D patients 69% (40/58) were treated with glucose lowering medication: 14 individuals were on insulin only, 20 on metformin only and 6 on insulin in combination with metformin; 31% (18/58) were treated with diet and physical activation only”.

- Same section: Last sentence of this section that begins with “Death occurs in 8%...”- needs grammatical attention. 

Response: The sentence has been grammatically corrected to:” Death occurred in 8% (42/505) (mortality rate 1.2/100 patient-years)”.

Re: Discussion section

Need more detail on the association of T2D and HLA B5701- compare with other literature. 

Response: This point is well taken. We now discuss the association of T2D and HLA B5701 in more depth. We have, however, failed to find any references on B 5701 in relation to T2D. So, there seems to be no studies of the relation between HLA B5701 and T2D in HIV. Following text has been inserted in the revised manuscript (Discussion):“The trend suggesting a possible association of T2D with HLA B5701 at baseline was surprising and difficult to explain as HLA B5701 has been found to be protective for T1D [19]. One explanation might be a survival bias since time with HIV-infection was also associated to T2D in univariate regression analyse at baseline and HLA B5701 is associated with restriction of HIV-replication in long-term non-progressors [21]. The HLA B5701 link needs further studying.”

The manuscript also been revised regarding spelling and grammatical errors by an English speaker.

---

## [Decision Letter · Decision Letter 1]

18 May 2021

PONE-D-20-38599R1

Development of type 2 diabetes and insulin resistance in people with HIV infection: Prevalence, incidence and associated factors

PLOS ONE

Dear Dr. Nystrom,

Thank you for submitting your manuscript to PLOS ONE. After careful consideration, we feel that it has merit but does not fully meet PLOS ONE’s publication criteria as it currently stands. Therefore, we invite you to submit a revised version of the manuscript that addresses the points raised during the review process.

There are a few remaining grammatical errors. Reviewer #1 has also a few additional minor concerns.

We look forward to receiving your revised manuscript.

Kind regards,

Graciela Andrei

Academic Editor

PLOS ONE

Journal Requirements:

Reviewers' comments:

Reviewer's Responses to Questions

**Comments to the Author**

1. If the authors have adequately addressed your comments raised in a previous round of review and you feel that this manuscript is now acceptable for publication, you may indicate that here to bypass the “Comments to the Author” section, enter your conflict of interest statement in the “Confidential to Editor” section, and submit your "Accept" recommendation.

Reviewer #1: (No Response)

Reviewer #2: All comments have been addressed

2. Is the manuscript technically sound, and do the data support the conclusions?

Reviewer #1: Yes

Reviewer #2: Yes

3. Has the statistical analysis been performed appropriately and rigorously? 

Reviewer #1: Yes

Reviewer #2: Yes

4. Have the authors made all data underlying the findings in their manuscript fully available?

Reviewer #1: Yes

Reviewer #2: Yes

5. Is the manuscript presented in an intelligible fashion and written in standard English?

Reviewer #1: Yes

Reviewer #2: Yes

6. Review Comments to the Author

Reviewer #1: Minor Concerns:

1. There are remaining grammatical errors, including missing periods, in the revised manuscript (outlined below). Also, review manuscript carefully with regards to using a comma or decimal point to separate numbers and be consistent throughout manuscript; in particular, tables 2 & 3 and throughout discussion.

Introduction: First sentence – Mortality and morbidity among people living with human immunodeficiency virus type 1 (PLHIV) has decreased over the last two decades.

Introduction: 4th Paragraph – Many studies have indicated that PLHIV on cART have an over risk for myocardial infarction, cerebrovascular events and type 2 diabetes (T2D) [6-11].

Materials and methods: Patients – Exclusion criteria were: having been diagnosed with T1D or T2D prior to the start of the study and refusing to participate.

Funding Statement: The study has received economical support from Physicians against AIDS research fund (Läkare mot AIDS forskningsfond).

Data availability: Participants in this study have not consented for their data to be used by other researchers.

Legends to figure: Figure 2. Proportions of T1D, T2D and insulin resistance at follow up. All PLHIV who developed T2D and insulin resistance in the prospectively followed cohort are included in this diagram.

2. Materials and methods:

Patients: Inclusion criteria number 2 needs to be corrected - 2. Born 2012 or earlier.

3. Results:

Development of T2D and insulin resistance at follow up – Second sentence reads “Another 7% (36/505) (M: n=19; F: n=17) developed insulin resistance (incidence of 1.0/100 patient-years).” However, table 1 indicates that 153/505 of the prospectively followed cohort already had insulin resistance at baseline. This would indicate that 36 of the remaining 352 individuals developed insulin resistance. Please clarify here and in discussion.

4. Consider adding sample sizes, n=570 and n=505, to your figure legends.

Reviewer #2: All concerns have been adequately addressed by the authors. I believe this manuscript is much stronger and acceptable for publication in this journal.

7. PLOS authors have the option to publish the peer review history of their article (what does this mean?). If published, this will include your full peer review and any attached files.

Reviewer #1: No

Reviewer #2: No

---

## [Author Response · Author response to Decision Letter 1]

26 May 2021

Responses to the reviewers’ specific comments:

Reviewer # 1:

Re: Minor concerns

The remaining grammatical errors including missing periods have been corrected. The use of comms and decimal point has been made consistent throughout the manuscript and in Tables II and III. 

Re: Introduction

First sentence: “Mortality and morbidity among people living with human immunodeficiency virus type 1 infection has decreased over the last two decades” has been changed to “Mortality and morbidity among people living with human immunodeficiency virus type 1 infection have decreased over the last two decades”

Re: Material and Methods

Patients: Inclusion criteria number 2 needs to be corrected

Response: Inclusion criteria number 2 has been clarified to: “Born before the 1st of January 2013”

Re: Results: 

1 Development of T2D and insulin resistance at follow up – Second sentence reads “Another 7% (35/505) (M:n=19; F:n=17) developed insulin resistance (incidence of 1.1/100 patient-years)”. However, table 1 indicates that 135/505 of the prospectively followed cohort already had insulin resistance at baseline. This would indicate that 36 of the remaining 352 individuals developed insulin resistance. Please clarify here and in discussion.

Response: We have made new calculations and rewritten the sentence to: “During a median follow-up time of 7.25 years (3485 patient-years) 9% (43/505) (M: n=43; F: n=0) developed T2D (incidence of 1.2/100 patient-years). Another 36 patients (M: n=19; F: n=17) developed insulin resistance (incidence of 1.0/100 patient-years). Thus, at follow up, 79 patients (M: n=62; F: n=17) had developed either T2D or insulin resistance (incidence of 2.3/100 patient-years). In total, the occurrence of either T2D or insulin resistance was 46% (232 (153+79)/505) at follow up, Figure 2”.

We also added the sentence: ”Insulin resistance occurred in 30% (153/505), Table 1” in the first part of the Results: Description of the Prospectively Followed Cohort for further clarification.

The Table 1 and Tables 2-3 have been slightly reorganized for clarification and improved symmetry as shown:

Table 1. Descriptive population data of the patients without diabetes (n=505) at baseline. 

 Patients

Total number (M; F, n (%)) 505 (M 440 (88%); F 65 (12%))

Median age in years (range) 57 (49-83)

HLA B 5701 pos n; (%) 21/500 (4%)

Median time with known HIV-infection 

in months (range) 204 (6-374)

AIDS diagnosis n; (%) 94 (19%)

Median initial, pre-ART, HIV-RNA

in copies/ml (range) 42100 (19-10000000)

Median CD4 nadir x106/L median (range) 193 (0-870)

CD4 nadir <200 x106/L n; (%) 266 (53%)

Median time in months without cART (range) 54 (0-372)

Treatment interruption of at least one month: prevalence in %; median time in months (range) 26%; 16.0 (1-105)

Ever on D4t (Stavudine) n; (%) 179 (35%)

Ever on DDI (Didanosine) n; (%) 179 (35%)

On cART n; (%) 491 (97%)

cART including a NNRTI n; (%) 306 (61%)

cART including a Protease inhibitor n; (%) 162 (32%)

cART including an Integrase inhibitor n; (%) 100 (20%)

cART including Emtricitabine n; (%) 260 (51%)

cART including Lamivudine n; (%) 193 (38%)

cART including Abacavir n; (%) 176 (35%)

cART including Tenofovir n; (%) 268 (53%)

cART including Zidovudine n; (%) 5 (1%)

Median CD4 count x106/L (range) 600 (30–1620)

HIV-RNA <100 copies/ml at baseline n; (%) 489 (97%)

Hypertension n; (%) 205 (41%)

Hyperlipidemia n; (%) 202 (40%)

Central obesity n; (%) 114 (23%)

Insulin resistance n; (%) 153 (30%)

On statin treatment n; (%) 115 (23%)

Table 2. Uni and multivariate analysis regarding T2D in all patients including patients with diabetes (n=570) at baseline. The multivariate analysis was controlled for age. All parameters with p<0.20 in the univariate analysis were entered in the multivariate equation. OR and 95% confidence interval.

 Univariate

OR (95% CI) P-value Multivariate

OR (95% CI) P-value

HLA B 5701 positive 2.6 (1.0-6.8) 0.046 2.6 (1.0-7.1) 0.055

HIV-months >the median (209 months) 1.9 (1.1-3.3) 0.03 ns

Ever an AIDS diagnosis 1.7 (1.0-3.1) 0.11 ns

First HIV-RNA >300000 copies/ml 1.4 (0.8-2.5) 0.21 

CD4 nadir <200 x 106/L 1.7 (1.0-3.0) 0.065 ns

CD4<200 x 106/L for >12 months 1.7 (1.0-3.1) 0.061 ns

Ever on D4t (Stavudine) 1.4 (0.8-2.4) 0.25 

Ever on DDI (Didanosine) 1.7 (1.0-2.9) 0.057 ns

Ever been a smoker 1.4 (0.8-2.4) 0.29 

Table 3. HIV-parameters, treatment, metabolic and inflammatory factors at baseline in relation to T2D development during follow up (n=505). Uni and multivariate analysis at baseline. The multivariate analysis was controlled for age. All parameters with p<0.20 in the univariate analysis were entered in the multivariate equation. Odds Ratio (OR) and 95% confidence interval.

 Univariate

OR (95% CI) P-value Multivariate

OR (95% CI) P-value

HLA B 5701 positive 1.6 (0.5-5.5) 0.46 

HIV-months >the median (209 months) 0.8 (0.4-1.5) 0.52 

Ever an AIDS diagnosis 0.8 (0.3-1.8) 0.59 

First HIV-RNA >300000 copies/ml 0.8 (0.4-1.6) 0.59 

CD4 nadir <200 x 106/L 1.4 (0.7-2.7) 0.31 

CD4<200 x 106/L for >12 months 0.9 (0.4-1.9) 0.76 

Ever on D4t (Stavudine) 0.9 (0.5-1.7) 0.88 

Ever on DDI (Didanosine) 1.2 (0.6-2.3) 0.60 

On an NNRTI 0.9 (0.5-1.6) 0.68 

On a protease inhibitor 1.2 (0.6-2.2) 0.68 

On an integrase inhibitor 0.6 (0.2-1.4) 0.22 

On abacavir 1.5 (0.8-2.7) 0.25 

On tenofovir 0.8 (0.4-1.4) 0.38 

Hypertension 1.9 (1.0-3.6) 0.051 ns

Triglycerides >2.6 mmol/L 3.6 (1.8-7.2) <0.001 2.4 (1.1-5.7) 0.036

Non-HDL cholesterol >3.8 mmol/L 1.5 (0.8-2.9) 0.20 ns

Central obesity 2.6 (1.3-5.2) 0.005 2.8 (1.2-5.7) 0.015

Insulin resistance 3.4 (1.8-6.5) <0.0001 2.4 (1.1-5.1) 0.018

Hs CRP >3.0 mg/L 1.6 (0.8-3.2) 0.20 ns

On statin treatment 2.6 (1.4-4.9) 0.004 2.6 (1.2-5.5) 0.015

Actual smoker 0.9 (0.5-1.8) 0.81 

Re: Consider adding sample sizes, n=570 and n=505 to your figure legends

Response: the figure legends have been changed to: “

Figure 1. Proportions of T1D, T2D and insulin resistance in the cross-sectional analysis at baseline (n=570).

Figure 2. Proportions of T1D, T2D and insulin resistance in the prospectively followed cohort at follow up (n=505).

Furthermore, the figure 2 has been corrected accordingly

Figure 1. Proportions of type 1 diabetes (T1D), type 2 diabetes (T2D) and insulin resistance in the cross-sectional analysis (n=570) at baseline

Figure 2. Proportions of type 2 diabetes (T2D) and insulin resistance in the prospectively followed cohort (n=505) at follow up.

---

## [Decision Letter · Decision Letter 2]

11 Jun 2021

PONE-D-20-38599R2

Development of type 2 diabetes and insulin resistance in people with HIV infection: Prevalence, incidence and associated factors

PLOS ONE

Dear Dr. Nystrom,

Thank you for submitting your manuscript to PLOS ONE. After careful consideration, we feel that it has merit but does not fully meet PLOS ONE’s publication criteria as it currently stands. Therefore, we invite you to submit a revised version of the manuscript that addresses the points raised during the review process.

Please revise your abstract, materials & methods, and results taking into account all the reviewer's comments.

We look forward to receiving your revised manuscript.

Kind regards,

Graciela Andrei

Academic Editor

PLOS ONE

Journal Requirements:

Reviewers' comments:

Reviewer's Responses to Questions

**Comments to the Author**

1. If the authors have adequately addressed your comments raised in a previous round of review and you feel that this manuscript is now acceptable for publication, you may indicate that here to bypass the “Comments to the Author” section, enter your conflict of interest statement in the “Confidential to Editor” section, and submit your "Accept" recommendation.

Reviewer #1: (No Response)

2. Is the manuscript technically sound, and do the data support the conclusions?

Reviewer #1: Yes

3. Has the statistical analysis been performed appropriately and rigorously? 

Reviewer #1: Yes

4. Have the authors made all data underlying the findings in their manuscript fully available?

Reviewer #1: Yes

5. Is the manuscript presented in an intelligible fashion and written in standard English?

Reviewer #1: Yes

6. Review Comments to the Author

Reviewer #1: Minor Concerns:

Re: Material and Methods

Inclusion criteria number 2 has been clarified to “Born before the 1st of January 2013.” However, this suggests that children as young as 8 years old could participate in the study. Please revise accordingly.

Re: Results:

The concerns regarding the incidence of insulin resistance at baseline in the prospective cohort were addressed but further clarity is needed. I would suggest the following changes addition to the second sentence of the paragraph “Another 36 patients, 10% (36/352) (M:n=19; F: n=17) developed insulin resistance (incidence of 1.0/100 patient-years).

In addition, the last sentence reads “In total, the occurrence of either T2D or insulin resistance was 46% (232 (153+79)/505) at follow up, Figure 2. I would suggest using ‘prevalence’ in place of ‘occurrence.’

Re: Abstract

Please make the correction above to the abstract as well since the old data for insulin resistance is still reflected.

Results: During follow up (3485 patient-years) 9% (43/505) developed T2D and 7% (34/505) insulin

resistance.

7. PLOS authors have the option to publish the peer review history of their article (what does this mean?). If published, this will include your full peer review and any attached files.

Reviewer #1: No

---

## [Author Response · Author response to Decision Letter 2]

14 Jun 2021

Responses to the minor comments:

Reviewer #1: Minor Concerns:

Re: Material and Methods

Inclusion criteria number 2 has been clarified to “Born before the 1st of January 2013.” However, this suggests that children as young as 8 years old could participate in the study. Please revise accordingly.

This is a typo, corrected to born before the 1st of January 1963 in the revised manuscript

Re: Results:

The concerns regarding the incidence of insulin resistance at baseline in the prospective cohort were addressed but further clarity is needed. I would suggest the following changes addition to the second sentence of the paragraph “Another 36 patients, 10% (36/352) (M:n=19; F: n=17) developed insulin resistance (incidence of 1.0/100 patient-years).

In addition, the last sentence reads “In total, the occurrence of either T2D or insulin resistance was 46% (232 (153+79)/505) at follow up, Figure 2. I would suggest using ‘prevalence’ in place of ‘occurrence.’

We appreciate this comment and have now change the text accordingly in the revised manuscript. The development of insulin resistance during follow up was 7% (36 out of 505)

Re: Abstract

Please make the correction above to the abstract as well since the old data for insulin resistance is still reflected.

Thanks for notice this, the abstract is now changed.

Results: During follow up (3485 patient-years) 9% (43/505) developed T2D and 7% (34/505) insulin resistance.

This is now corrected in the Abstract (Results). “During follow up (3485 patient-years) 9% (43/505) developed T2D and 7% (36/505) insulin resistance”

---

## [Editor Report · Decision Letter 3]

21 Jun 2021

Development of type 2 diabetes and insulin resistance in people with HIV infection: Prevalence, incidence and associated factors

PONE-D-20-38599R3

Dear Dr. Nystrom,

We’re pleased to inform you that your manuscript has been judged scientifically suitable for publication and will be formally accepted for publication once it meets all outstanding technical requirements.

Kind regards,

Graciela Andrei

Academic Editor

PLOS ONE
---

## [Editor Report · Acceptance letter]

22 Jun 2021

PONE-D-20-38599R3 

Development of type 2 diabetes and insulin resistance in people with HIV infection: Prevalence, incidence and associated factors 

Dear Dr. Nyström:

I'm pleased to inform you that your manuscript has been deemed suitable for publication in PLOS ONE. Congratulations! Your manuscript is now with our production department. 

Kind regards, 

on behalf of

Dr. Graciela Andrei 

Academic Editor

PLOS ONE